# Digital Tomosynthesis: Review of Current Literature and Its Impact on Diagnostic Bronchoscopy

**DOI:** 10.3390/diagnostics13152580

**Published:** 2023-08-03

**Authors:** Anant Jain, Adrish Sarkar, Shaikh Muhammad Noor Husnain, Brian Cody Adkinson, Ali Sadoughi, Abhishek Sarkar

**Affiliations:** 1Department of Pulmonary, Critical Care, and Sleep Medicine, Westchester Medical Center, New York Medical College, Valhalla, NY 10595, USA; anant.jain@wmchealth.org (A.J.); shaikh.husnain@wmchealth.org (S.M.N.H.); 2Department of Radiology, Nassau University Medical Center, East Meadow, NY 11554, USA; asarkar@numc.edu; 3Department of Pulmonary, Critical Care, and Sleep Medicine, Miller School of Medicine, Jackson Memorial Hospital, University of Miami, Miami, FL 33136, USA; cody.adkinson@miami.edu; 4Department of Pulmonary Medicine, Albert Einstein College of Medicine, Montefiore Medical Center, Bronx, NY 10467, USA; asadough@montefiore.org

**Keywords:** digital tomosynthesis, lung nodules, bronchoscopy, cone beam CT, robotic bronchoscopy

## Abstract

Bronchoscopy has garnered increased popularity in the biopsy of peripheral lung lesions. The development of navigational guided bronchoscopy systems along with radial endobronchial ultrasound (REBUS) allows clinicians to access and sample peripheral lesions. The development of robotic bronchoscopy improved localization of targets and diagnostic accuracy. Despite such technological advancements, published diagnostic yield remains lower compared to computer tomography (CT)-guided biopsy. The discordance between the real-time location of peripheral lesions and anticipated location from preplanned navigation software is often cited as the main variable impacting accurate biopsies. The utilization of cone beam CT (CBCT) with navigation-based bronchoscopy has been shown to assist with localizing targets in real-time and improving biopsy success. The resources, costs, and radiation associated with CBCT remains a hindrance in its wider adoption. Recently, digital tomosynthesis (DT) platforms have been developed as an alternative for real-time imaging guidance in peripheral lung lesions. In North America, there are several commercial platforms with distinct features and adaptation of DT. Early studies show the potential improvement in peripheral lesion sampling with DT. Despite the results of early observational studies, the true impact of DT-based imaging devices for peripheral lesion sampling cannot be determined without further prospective randomized trials and meta-analyses.

## 1. Introduction

Pulmonary nodules are single, well-circumscribed radiographic opacities up to 30 mm in diameter, surrounded by normal lung parenchyma and without associated atelectasis, hilar enlargement, or pleural effusion [1]. They are detected in about 1.5 million people annually [2]. Lung cancer, the leading cause of cancer deaths worldwide with 1.8 million deaths in 2020 [3], is the main differential diagnosis of pulmonary nodules. However, 66% of lung cancers are still diagnosed at a late stage [4]. Therefore, accurate assessment and management of pulmonary nodules are crucial. The evaluation usually starts with a lung cancer risk prediction calculator to estimate the probability of malignancy [5]. Depending on the risk score, management options include serial CT follow-up, PET-CT, nonsurgical biopsy, or surgical resection [6]. Most nodules are in the intermediate risk category, which may require biopsy. Nonsurgical biopsy methods include CT-guided transthoracic needle biopsy (TTNB) or bronchoscopic biopsy. TTNB has a high diagnostic yield (67–97%) [7] but is limited to peripheral lesions and has a high complication rate. Conventional bronchoscopy is safe but has a low diagnostic yield (14–31%) [8] and is limited to central lesions. Guided bronchoscopy aims to overcome the limitations of conventional bronchoscopy by providing a safe and effective procedure for peripheral lesions. It consists of navigational bronchoscopy techniques such as electromagnetic navigational bronchoscopy (ENB), radial endobronchial ultrasound (REBUS) with ultrathin bronchoscopy, and virtual bronchoscopy (VB).

## 2. Electromagnetic Navigational Bronchoscopy (ENB)

In 2006, the first prospective, controlled clinical study on ENB in humans was reported for use in peripheral lesion biopsy. Since then, navigational technologies have advanced and enabled the creation of a 3D map of the airways from CT scan images. This map provides a virtual bronchoscopy with a bronchoscopic view and a pathway through the airway lumen. Moreover, EM tracking facilitates real-time positional guidance and directional cues. The superDimension system^TM^ (Medtronic, Minneapolis, MN, USA) and the Veran Thoracic Navigation System^TM^ (Veran Medical Technologies Inc, St Louis, MO, USA) are the most used EM navigational bronchoscopy (ENB) systems [9].

The NAVIGATE study was a prospective, multi-center cohort study that evaluated the diagnostic performance and safety of electromagnetic navigation bronchoscopy (ENB) for peripheral pulmonary lesions. The study enrolled 1215 subjects with suspicious lung nodules. In the sample, 94% of patients who underwent ENB-guided biopsy had navigation completed and tissue obtained. The 12-month diagnostic yield was 73%. The study also reported a low incidence of ENB-related adverse events (grade 2 or higher), such as pneumothorax (2.9%), hemorrhage (1.5%), and respiratory failure (0.7%) [10].

## 3. Radial EBUS

Radial EBUS (REBUS) uses a rotating ultrasound mini-probe to confirm the location of peripheral lesions before biopsy. The probe was initially developed in Japan and was modeled after larger radial probes used to evaluate duodenal lesions [11]. The probe produces a circumferential ultrasound image and is inserted into bronchial subsegments until a signal shows a peripheral lesion. The views of peripheral lesions are “concentric” or “eccentric” depending on how the probe is surrounded by the lesion. The overall diagnostic yield of REBUS is 70% [12]. The factors that affect the yield are lesion size, nature, bronchus sign, and probe position. A larger size, malignant nature, positive bronchus sign, and probe within the lesion increase the yield [10]. In another study, the diagnostic yield was 84% when the probe was within the lesion and 48% when it was adjacent to the lesion [13].

## 4. Robotic Bronchoscopy

Robotic bronchoscopy (RB) is a recent advancement in technology aiding lung nodule biopsy. RB uses a robotic arm to guide a catheter with a camera, light, and biopsy tools through the patient’s airways to access and biopsy nodules in the hard-to-reach periphery of the lungs. There are two major platforms commonly utilized in North America. The Monarch^TM^ robotic flexible endoscopy platform (Auris Heath, Redwood City, CA, USA) uses a bronchoscope and a sheath that can bend in different ways and be controlled by separate robotic arms. The sheath covers the scope, which has a camera and a channel for tools. The platform also has a tower with a screen and a controller. The platform uses electromagnetic navigation with sensors on the chest, and it can work with a C-arm or CBCT. The screen can show vision, navigation, REBUS, and CT overlay together, and it is operated with a controller that has two thumb-sticks. The inner scope has a 4.2 mm outer diameter. Compared to its competing robotic platform, the Monarch^TM^ system has continuous airway visualization while performing biopsies in the periphery. The Ion^TM^ endoluminal system (Intuitive Surgical, Sunnyvale, CA, USA) is the second RB platform widely available in North America. Unlike the Monarch^TM^ system, Ion^TM^ uses a proprietary shape-sensing technology through the length of the bronchoscope to track position and navigation. No electromagnetic navigation sensors are needed. The bronchoscope has a channel for tools, but the vision probe containing the camera must be removed during biopsy of lesions. The outer diameter is 3.5 mm [14]. A comparison of overall features of each platform can be found in Table 1.

## 5. Clinical Data Using Robotic Bronchoscopy

Investigators at the University of Chicago conducted one of the initial multi-center, retrospective studies. They noted that RB had a diagnostic yield ranging from 69.1% to 77%, depending upon whether inflammatory findings were included in criteria for diagnosis [15]. Similarly, investigators at Memorial Sloan Kettering Cancer Center published a retrospective analysis on 151 peripheral nodule biopsies using RB (Ion^TM^). The overall diagnostic yield was noted to be 81.7% [16]. A comparative study was recently published by researchers at Mayo Clinic comparing yield and safety of robotic bronchoscopy and CT-guided trans-thoracic needle biopsy. Investigators noted similar diagnostic yields of 87.6% for RB and 88.4% for CT-guided TTNB when assessing malignant disease. RB had a lower complication rate of 4.4% compared to 17% for TTNB [17]. Despite early observation data, prospective randomized trials with head-to-head comparison of RB to ENB with REBUS are necessary to truly differentiate if a difference in diagnostic yield exists.

## 6. Variables Impacting Diagnostic Yield

Despite several technological advances, clinicians have noted that CT body divergence and atelectasis are major variables impacting further improvement in diagnostic yield [18]. Guided bronchoscopy systems, whether robotic or virtual navigation, use a virtual map of the patient’s airways based on a pre-procedural computed tomography (CT) scan to locate target lesions. However, this approach is prone to errors due to the discrepancy between the expected and actual lesion location caused by changes in lung anatomy. This phenomenon, known as “CT-to-body divergence”, lowers the diagnostic yield, prolongs the procedure time, and poses a challenge for the operator. Several factors contribute to CT-to-body divergence in guided bronchoscopy such as lung volume differences, time gap between CT and bronchoscopy, and other factors such as interim variations in nodule size, mucus plugging, and pleural effusions that can cause anatomical distortion [18]. CT-to-body divergence can affect any platform that relies only on a pre-procedural CT for guidance.

Similarly, atelectasis is a common problem impacting overall diagnostic yield. Atelectasis can occur within minutes of general anesthesia induction and most often involves the lower lobes. The causes of atelectasis include prolonged intubation time, suboptimal ventilation protocols, high fractions of inspired oxygen leading to absorption atelectasis, and distal wedging of the bronchoscope [19]. Atelectasis and anatomical changes in the airways can evolve during the bronchoscopic procedure, resulting in a dynamic airway structure that does not match the original virtual map. Atelectasis reduces the distance between the lesion and the pleura, increasing the risk of pneumothorax from instrumentation. Atelectatic lung can also mimic lung lesions on radial EBUS due to its increased density compared to aerated lung [19].

The following are some recommendations for ventilator management before bronchoscopy [19]:Pre-procedural incentive spirometry can help recruit lung volume and prevent atelectasis [20].The use of 100% oxygen during pre-oxygenation can induce absorption atelectasis, so the lowest tolerable FiO_2_ should be used [21].Lengthy intubation times may increase the risk of atelectasis. General anesthesia using total intravenous anesthesia (TIVA) with propofol and muscle paralysis is optimal [22].Application of PEEP throughout induction can also prevent atelectasis [22].Higher PEEP with the lowest tolerable FiO_2_ as guided by oxygen saturation should be maintained. PEEP of up to 10–12 cm H_2_O may be beneficial for upper lobe biopsies, and higher PEEP may be needed for lower lobe biopsies [22].Recruitment maneuvers immediately after intubation can reverse any intubation atelectasis. This is especially important if intubation was difficult or prolonged [21].

## 7. Cone Beam CT

The discrepancy in target localization during bronchoscopy has led to investigation into imaging solutions providing real-time, multi-axis visualization of lesions. In recent years, cone beam computer tomography (CBCT) has been utilized for confirmation of peripheral lesion targets. CBCT differs from conventional computer tomography (CT) through its use of a cone-shaped X-ray beam that projects onto a flat detector sensing two dimensions. Conventional CT uses fan-shaped X-ray beams that project onto multiple, single dimension detectors over multiple rotations. Whereas conventional CT reconstructs volumetric data from slice-by-slice images in the axial plane, CBCT reconstructs volumetric data for a specific anatomical field of view. The differences in image acquisition allows CBCT to obtain similar volumetric data with a single rotation and shorter duration [1]. Current CBCT has similar image quality when compared to conventional CT but with reduced contrast ratio. Such images allow one to determine the location of a bronchoscopic biopsy tool in relation to a lesion’s location (Figure 1) [23].

In early studies using CBCT and peripheral nodule sampling, the visualization of a tool within a lesion was determined to have the strongest impact on diagnostic yield [24]. Additionally, many cone beam platforms allow for augmented fluoroscopy, where a 3D image of the lesion is overlaid onto a 2D real-time fluoroscopic image to allow for sampling guidance [25]. The calculated radiation dose for CBCT varies widely depending on model, duration of scan, and targeted field size [26]. Additionally, the method of reporting radiation dose differs between CBCT and conventional CT [27]. However, CBCT overall has less radiation compared to conventional CT, but similar radiation dose compared to reported doses for lung cancer screening CT [27,28,29]. There are many different types of CBCT scanners. Fixed systems include ceiling, floor, biplane, and robotic. Fixed systems require large, dedicated rooms to accommodate both the CBCT, general anesthesia machines, and procedural devices. Newer mobile and compact CBCT have recently been developed and will be discussed later in this article. 

Many studies have emerged investigating CBCT as a means of increasing diagnostic yield and detecting CT to body divergence for peripheral nodule sampling. One of the earliest retrospective studies combined ultra-thin bronchoscopy with navigation bronchoscopy and CBCT in 29 patients and reported a diagnostic yield of 92% for malignant lesions and 86.7% in benign lesions [30]. Around the same time, a prospective pilot study in North America investigated the impact of CBCT on diagnostic yield using ultra-thin bronchoscopy and REBUS for peripheral nodules in 20 patients. The investigators reported a diagnostic yield of 70% compared to 50% prior to incorporation of CBCT [31]. A similar retrospective study using navigation bronchoscopy before and after CBCT incorporation in 62 total patients showed a diagnostic yield of 74.2% when combining CBCT with navigation bronchoscopy compared to 51.6% for navigation bronchoscopy alone [32]. A larger scale retrospective study in 75 patients combining navigation bronchoscopy with CBCT-associated augmented fluoroscopy functionality reported a combined diagnostic yield of 83% [33].

As robotic bronchoscopy gained popularity in sampling peripheral lesions, the impact of robotic bronchoscopy with CBCT began to be investigated. In a retrospective study of 52 patients, the combination of robotic bronchoscopy with CBCT demonstrated a diagnostic yield of 86% [34]. A larger scale retrospective analysis combining robotic bronchoscopy with CBCT for 200 biopsies demonstrated a diagnostic yield of 91.4% [35]. Overall, the incorporation of CBCT in peripheral nodule biopsy has shown an overall increase in diagnostic yield regardless of bronchoscopy platform. However, certain cautions must be taken. The majority of CBCT bronchoscopy studies are observational and single center. Definitions of diagnostic yield vary widely between studies in terms of follow-up duration and histopathological findings. Additionally, the use of dedicated CBCT, bronchoscopy platforms, ventilators, and other tools presents significant financial and resource burdens for medical institutions. These factors limit adoption of peripheral sampling combining bronchoscopy with CBCT.

## 8. Digital Tomosynthesis-Based Imaging Devices

Due to the resource limitations of CBCT, digital tomosynthesis (DT) has recently gained increased popularity. Digital tomosynthesis utilizes computer-based reconstruction algorithms to create radiographic images with depth of field from multiple single-plane X-ray images. The images are obtained over a limited range of angles with an X-ray tube and detector circling around an object [36]. The concepts of tomosynthesis are largely credited to the geometric tomography theories and early devices of Ziedses des Plantes [37]. The advent of computer processing improved upon geometric tomosynthesis by utilizing processing algorithms that reduced the blur effects of synthesized images [38,39]. Unlike conventional CT where images are obtained over 180 to 360 degrees, digital tomosynthesis utilizes X-ray images obtained over angles as small as 50 degrees [40]. Thus, the cumulative radiation dose of DT, while potentially slightly higher than two-view X-ray images, is still a fraction of the radiation used in computed tomography or CBCT [41]. Similarly, the limited range of motion allows for smaller procedural suites and the ability to use conventional devices, such as a C-arm fluoroscope, with computer processing adjuncts [42,43]. However, the image resolution compared to CBCT is significantly lower in DT. Structures in the center of rotation have less motion artifact and are more readily discernible compared to structures in the periphery. Consequently, the spatial relation of structures can only be distinguished for structures in the center of a DT image’s field of view. Although limited from a diagnostic imaging perspective, DT’s capabilities allow for effective procedural use by distinguishing the spatial relationship of small lesions with biopsy tools.

## 9. Illumisite^TM^

Currently, in North America, there are several medical device manufacturers incorporating digital tomosynthesis for peripheral nodule sampling, each with different methods of implementation. Medtronic (Minneapolis, MN, USA) recently changed their electromagnetic navigation platform (superDimension^TM^) to incorporate digital tomosynthesis to correct the real-time positioning of the peripheral target during navigation (Illumisite^TM^). The updated device replaces the superDimension navigation console. It attaches to traditional X-ray fluoroscopy C-arm (see manufacturer for requirements). After an initial navigation to the target is performed using a preplanned pathway derived from a prior CT scan, a digital tomosynthesis image is captured through rotating the C-arm, either manually or through motorized mechanisms. Once the nodule is identified on imaging through software programing and clinician identification, the navigation pathway is updated to reflect the real-time nodule’s position (Figure 2). In its current iteration, the platform does not provide multi-plane images or three-dimensional images showcasing the location of the target nodule and the catheter. Additionally, the platform’s navigation adjustment is only compatible with the Illumisite^TM^ navigation platform and no other manufacturers.

## 10. LungVision^TM^

Another platform incorporating digital tomosynthesis is LungVision^TM^ (Ramat Ha Sharon, Israel). Similar to Illumisite^TM^, the platform consists of a console tower that connects to conventional X-ray fluoroscopy C-arm (see manufacturer requirements) to create digital tomosynthesis images. The platform uses fluoroscopic navigation instead of electromagnetic navigation. Similar to the Illumisite^TM^ platform, once the target nodule is navigated to using LungVision^TM^ fluoroscopic navigation or other navigation platforms, a digital tomosynthesis image is captured by rotating the fluoroscopic C-arm, either manually or through motorized systems. Through proprietary software and algorithms, the platform can correct the location of the target within the fluoroscopic navigation to the real-time location. The navigation correction cannot be applied to other manufacturer platforms. Unlike the Illumisite^TM^ device, LungVision^TM^ can also provide multi-plane and three-dimensional images showing the relationship between the nodule and biopsy tools. Due to the limitations of digital tomosynthesis, it is difficult to discern soft tissue structures peripheral to the nodule when compared to images provided by CBCT. However, the images relay sufficient information to correlate the position of biopsy tools in relation to the nodule itself. Many clinicians utilize such images depicting the tool and nodule relationship with separate navigation platforms of their choosing.

## 11. Robotic Bronchoscopy with Digital Tomosynthesis and Electromagnetic Navigation

In the realm of robotic bronchoscopy, Noah Medical developed a robotic bronchoscopy platform that utilizes digital tomosynthesis called The Galaxy System^TM^. The device uses a bronchoscope smaller (4 mm) than The Monarch^TM^ platform (4.2 mm), but larger than the IonTM Endoluminal System (3.5 mm). Similar to the The Monarch^TM^ platform, The Galaxy System^TM^ provides real-time bronchoscopic visualization. The Galaxy System^TM^ uses a proprietary combination of digital tomosynthesis and EMN platforms to aid both navigation and real-time biopsy localization (TiLT Technology^TM^). The Galaxy System recently received FDA approval. However, there are currently no human studies utilizing the platform and current studies are limited to target localization in animal studies [44].

## 12. Mobile Cone Beam

Due to the resource and cost restrictions of fixed CBCT, several manufacturers developed mobile CBCT platforms that utilize the concepts of X-ray based tomosynthesis. Siemens Healthineers (Erlangen, Germany) developed a mobile CBCT known as Cios 3D Spin Mobile (Cios Spin) that is slightly larger in dimensions than a conventional X-ray C-arm Fluoroscope. The Cios Spin rotates on a single axis similar to a conventional X-ray C-arm. With the use of complementary metal oxide semiconductor (CMOS) for sensors and modern computational reconstruction, the device is capable of producing images similar to conventional fixed CBCT [45]. The device is also capable of creating traditional 2D X-ray fluoroscopic images as well as 3D reconstructions. The 3D image is captured with a 30 s motorized C-Arm spin. Due to size limitations of the sensors and X-ray generators, the field of view of mobile CBCT platforms is smaller than that of stationary CBCT (volume size 16 × 16 × 16 cm at 512 × 512 × 512 pixels) (Figure 3). As an imaging platform, the Cios Spin is able to provide the bronchoscopist with 3D images showing the relationship of biopsy tools with target lesions. Unlike X-ray based digital tomosynthesis platforms, it is capable of showing finer soft tissue details. The Cios Spin also has the ability to correct the navigation target to the real-time location for the Ion^TM^ Endoluminal System robotic bronchoscopy. GE Healthcare has also developed a mobile CBCT known as OEC 3D. The device is similar to the CIOS Spin in that it provides multi-plane CBCT images as well as 3D reconstructions defining spatial relationships of biopsy tools and target lesions. In its current iteration, the platform has not been integrated into other navigation bronchoscopy devices to allow for target adjustment based on real-time location. A summary comparison of the features of each digital tomosynthesis-based platform can be found in Table 2.

## 13. Clinical Data for Illumisite^TM^

Since the adoption of digital tomosynthesis for peripheral nodule sampling, there have been several published studies with each particular device. Using the Illumisite^TM^ platform, several institutions performed observation studies comparing diagnostic yield before and after implementation of digital tomosynthesis to electromagnetic navigation bronchoscopy (ENB). Researchers at Vanderbilt University, using a conservative definition of diagnostic yield, noted 79% yield using ENB with digital tomosynthesis compared to 54% using EMN alone [46]. Clinicians at the Brody School of Medicine published a retrospective analysis of diagnostic yield of peripheral nodules biopsy using ENB with digital tomosynthesis in 72 patients and noted an overall yield of 87% [47]. The group previously published an overall diagnostic yield of 73.6% in a 2015 retrospective study [48]. Recently, researchers at Vanderbilt University performed one of the few studies comparing different navigation platforms. The group conducted a retrospective comparative study comparing diagnostic yield in 133 patients using ENB combined with digital tomosynthesis to the yield in 170 patients using the shape-sensing robotic assisted bronchoscopy platform (Ion^TM^). Diagnostic yield of the robotic platform was 77% compared to 80% for ENB with digital tomosynthesis device. When accounting for nodule size, location of the lesion, presence of bronchus sign, and sex, no statistically significant difference was noted [49]. At the time of this review article, the group is conducting a randomized control study to compare the diagnostic yield of the two platforms (RELIANT—clinical trial identifier NCT05705544). Despite findings from early observation studies, comparisons of diagnostic yield for ENB augmented by DT with prior bronchoscopy platforms cannot be concluded without more prospective, randomized trials. 

## 14. Clinical Data for LungVision^TM^

Since the development of the LungVision^TM^ fluoroscopic and digital tomosynthesis navigation platform in 2016, there have been few clinical-based studies. Researchers in Israel performed a single-center retrospective study on 63 biopsies from 2016 to 2020 assessing safety and diagnostic yield. The researchers noted a diagnostic yield of 82%. The yield was smaller for lesions less than 20 mm in size (72%). Notably, the diagnostic yield progressively improved with familiarity using the device and with use of a second generation of the device, from 67% in 2016 to 82% in 2020 [50]. Researchers in North America published a prospective, multi-center study utilizing the LungVision platform for both navigation and biopsy. They noted an overall diagnostic rate of 75% in 55 patients over 1 year [51]. As described previously, the LungVision^TM^ platform is also capable of producing standalone multi-plane images depicting the relationship of biopsy tools with peripheral lesions. Researchers at the University of Chicago who utilized these capabilities with RB noted an immediate diagnostic yield of 84% in a retrospective review of 45 patients [52]. Current data using LungVision^TM^ remain limited to single-institution, small-scale studies. Further multi-center, large-scale comparison studies are necessary to gauge diagnostic accuracy compared to previous bronchoscopy methods.

## 15. Clinical Data for CIOS Spin

Researchers at Sloan Kettering Cancer Center published a feasibility case series using robotic bronchoscopy combined with CIOS Spin and noted a 90% tool in lesion rate. Diagnostic yield was not reported [53]. This was followed up by a larger scale retrospective analysis of 131 robotic bronchoscopy procedures where an 81% diagnostic yield was reported. However, the number of cases in which CIOS Spin was specifically utilized was not delineated [16]. Researchers at Mayo Clinic investigated whether combining robotic bronchoscopy with CIOS Spin can account for divergence and help improve diagnostic yield in a single-center prospective study. For 30 nodules that were sampled, the team noted divergence in 50% of cases. Divergence was defined by greater than 10 mm distance between the pre-procedure CT location and the location of the nodule during the case based on CIOS Spin. The overall diagnostic yield was noted to be 93% [54]. CIOS Spin has also been utilized in studies in platforms outside of RB. Investigators at the Albert Einstein College of Medicine initially reported their experience with thin bronchoscopy combined with CIOS Spin in a case series [55]. Researchers at MD Anderson Cancer Center published a retrospective, single-center study evaluating diagnostic yield utilizing thin or ultra-thin bronchoscopes, radial EBUS, and the CIOS Spin. Using a conservative definition for diagnostic yield, the researchers noted an overall diagnostic yield of 78% in 51 patients [56]. Similar to other DT-based platforms, CIOS Spin has not been incorporated into meta-analyses, limiting the comparison of diagnostic yield with other bronchoscopic technologies (Table 3).

## 16. Summary

There have been rapid technological advancements in peripheral lung nodule biopsy techniques over the last several years. Despite such advancements, issues pertaining to CT divergence and atelectasis remain. The incorporation of digital tomosynthesis-based imaging modalities allows bronchoscopists to localize a peripheral target more effectively and easily in real time. The ability to visualize biopsy tools within a lesion through certain platforms provides bronchoscopists greater confidence in results for benign lesions. However, further multi-center, randomized trials are warranted to truly delineate changes in diagnostic yield through digital tomosynthesis. Cost-effectiveness studies are also necessary to determine feasibility for widespread utilization of these imaging modalities. The addition of imaging modalities on top of navigation platform significantly increases the cost of sampling peripheral lesions. Clear, consistent definitions of “diagnostic yield” such as delineating what constitutes a “benign disease diagnosis” and distinguishing immediate diagnosis over long-term diagnosis are warranted. The impact of digital tomosynthesis-based imaging platforms holds promise, but current clinical research data remain within the early stages.

## Figures and Tables

**Figure 1 diagnostics-13-02580-f001:**
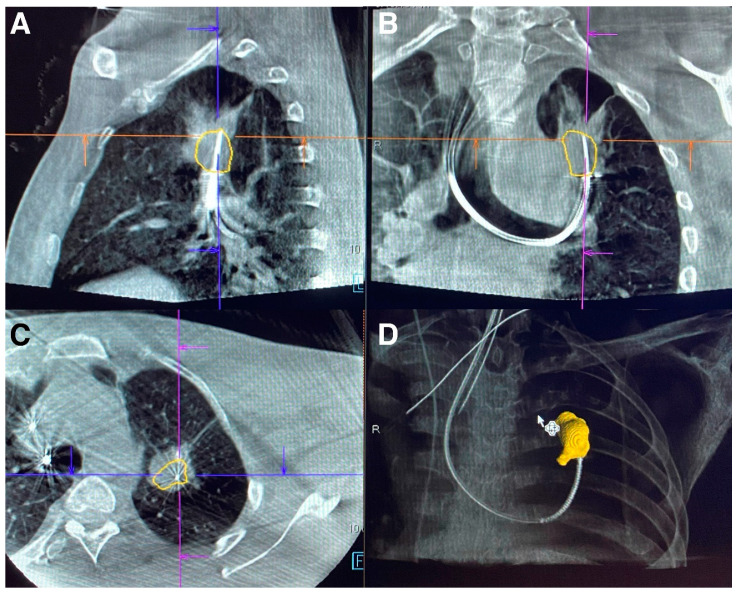
Multi-plane images from fixed CBCT of lung mass with biopsy needle in lesion using robotic bronchoscopy. (**A**) Sagittal section. (**B**) Coronal section. (**C**) Axial section. (**D**) 3D reconstruction. Images Courtesy of Authors.

**Figure 2 diagnostics-13-02580-f002:**
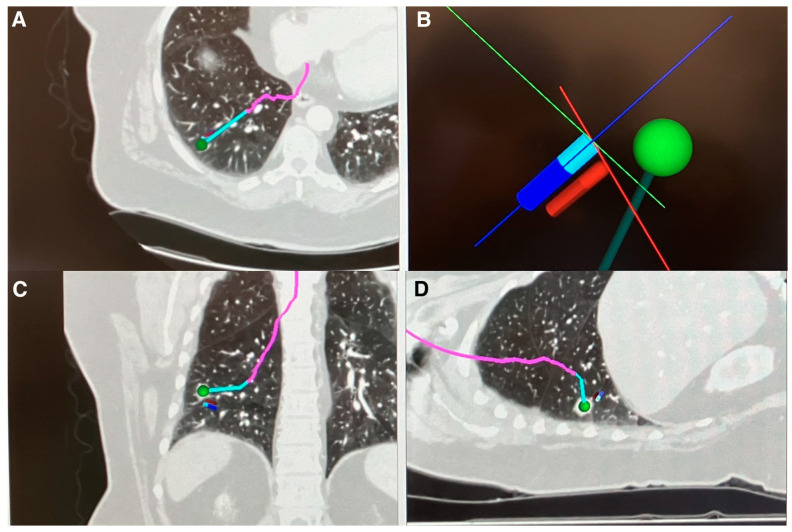
Illumisite^TM^ navigation pathway after correcting for real-time location of nodule using digital tomosynthesis. (**A**) Axial CT view with corrected navigation pathway (pink and blue line). (**B**) 3D Map Dynamic View showing corrected pathway (dark green line) based on nodule (green dot) location extracted through digital tomosynthesis next to current catheter position from originally planned pathway (blue stick). (**C**) Coronal CT view with corrected navigation pathway (pink and blue line). (**D**) Sagittal CT view with corrected navigation pathway (pink and blue line). Images Courtesy of Authors.

**Figure 3 diagnostics-13-02580-f003:**
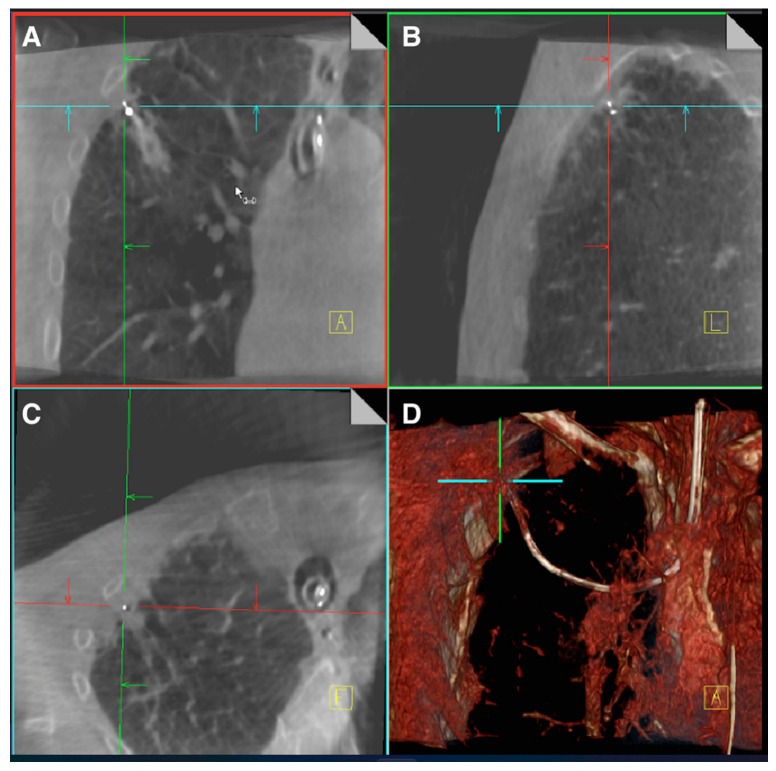
CIOS Spin^TM^ images of a RUL nodule biopsied with robotic bronchoscopy. (**A**) Coronal view. (**B**) Sagittal view. (**C**) Axial view. (**D**) 3D reconstruction. Images Courtesy of Authors.

**Table 1 diagnostics-13-02580-t001:** Comparison of ceatures of the Monarch^TM^ and Ion^TM^ robotic bronchoscopy devices.

Attributes	Monarch^TM^ RB Platform	Ion^TM^ RB System
Bronchoscope	Articulating bronchoscope within an articulating sheath	Single ultrathin bronchoscope with integrated shape-sensing technology
Diameter	6.0 mm (sheath), 4.4 mm (scope)	3.5 mm
Working channel	2.1 mm	2 mm
Vision probe	Integrated camera	Removable vision probe
Articulation range	Up to 130 degrees (sheath), up to 180 degrees (scope)	Up to 180 degrees
Control mechanism	Video-game-style controller with two thumb-sticks	Ball mouse and scroll wheel
Navigation system	Electromagnetic navigation with sensors on chest	Shape-sensing technology
Monitor display	Vision, navigation, REBUS, and CT overlay	Navigation, fluoroscopy, virtual overlay, and either vision or REBUS
Cost	Above mid six-figure USD	Above mid six-figure USD

**Table 2 diagnostics-13-02580-t002:** Comparison of features of digital tomosynthesis-based imaging devices for peripheral nodule biopsy.

Relative Cost †	Provides 3D Reconstruction Images *	Incorporates into Conventional C-Arm Based Fluoroscope *	Corrects Navigation Pathway Based on Realtime Nodule Positioning *	Provides Multi-Plane Images *	Device
Low six-figure, USD	No	Yes	Yes—Illumisite Platform only	No	Illumisite^TM^
Mid to high five-figure, USD	Yes	Yes	Yes—LungVision Platform Only	Yes	LungVision^TM^
Mid six-figure, USD	Yes	No—Standalone C-Arm with CMOS sensor to provide CBCT images	Yes—Ion Robotic Bronchoscopy Platform Only	Yes	CIOS Spin

* Features subject to change. Contact manufacturers for updated features. † Based on author’s experience. Contact manufacturers for official pricing.

**Table 3 diagnostics-13-02580-t003:** Comparison of diagnostic yields and complications between digital tomosynthesis-based imaging devices.

Platform	* Diagnostic Yield %	Study Type	Complications
Illumisite^TM^	79%	Retrospective comparative study between ENB with digital tomosynthesis (*n* = 67) vs. standard ENB (*n* = 100) [46]	Pneumothorax 1.5%
Illumisite^TM^	87%	Retrospective, single-center review study [47]	Pneumothorax 2.5%
Illumisite^TM^	80%	Retrospective comparative study; ENB combined with digital tomosynthesis (n = 133) vs. shape sensing robotic bronchoscopy (*n* = 170) [49]	Pneumothorax 1.8%
LungVision^TM^	77.8% (73–82%)	Retrospective, single-center study (*n* = 63) [50]	Pneumothorax 1.6%
LungVision^TM^	75%	Prospective, multi-center study (*n* = 55) [51]	None reported
LungVision^TM^	84%	Retrospective, single-center study (*n* = 45) [52]	Pneumothorax 8%
CIOS Spin	Not reported (Tool in Lesion 90%)	Feasibility case series (*n* = 10) [53]	None reported
CIOS Spin	93%	Prospective, single-center study (*n* = 30) [54]	None reported
CIOS Spin	100%	Case series (*n* = 4) [55]	None reported
CIOS Spin	78%	Retrospective, single-center study (*n* = 51) [56]	Pneumothorax 3.3%

* Diagnostic yield is from observation data and should not be compared to previous meta-analysis diagnostic yield for bronchoscopy without digital tomosynthesis-based technology.

## Data Availability

Not applicable.

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
