# Peer review of "Digital Tomosynthesis: Review of Current Literature and Its Impact on Diagnostic Bronchoscopy"

_diagnostics, 2023, doi:10.3390/diagnostics13152580_

Round 1

Reviewer 1 Report

In general, the review is well written regarding format and scientific aspects

My comments:

1. Meta-analyses and systematic reviews have shown that the diagnostic yield of bronchoscopy is around 70% no matter which method is used (radial-probe EBUS, navigational techniques, robotic bronchoscopy, thin bronchoscopy, etc.).

Digital tomosynthesis is a new technique in the field of periheral lung lesions. The number of studies on this method are not many and thus, the high yields obtained in these several studies cannot be generalized. Moreover, there are no systematic reviews/meta-analyses yet.

Thus, the authors should mention about and emphasize the issues above in appropriate parts of the text and also in the summary part:

2. Cost or cost-effectiveness of all methods mentioned should be given in the text and mentioned in the summary part.

3. There should be a table giving the diagnostic yields, complication rates and cost / cost-effectiveness of all the methods mentioned

4. The summary part should be re-written as an overview and comment on all mentioned in the review with a perspective of real life experience.

Author Response

1. Meta-analyses and systematic reviews have shown that the diagnostic yield of bronchoscopy is around 70% no matter which method is used (radial-probe EBUS, navigational techniques, robotic bronchoscopy, thin bronchoscopy, etc.).

Digital tomosynthesis is a new technique in the field of periheral lung lesions. The number of studies on this method are not many and thus, the high yields obtained in these several studies cannot be generalized. Moreover, there are no systematic reviews/meta-analyses yet.

Thus, the authors should mention about and emphasize the issues above in appropriate parts of the text and also in the summary part:

Line 27-30 – abstract changed to convey cannot equate observational data with previously meta-analysis

Line 113-115 – line reflecting RB needs to be compared to EMN -REBUS head to head prospectively to truly compare yields

Line 339-342- DT yields cannot be compared without prospective, higher quality data

Line 357-359- LungVision studies are very limited to small scale, single center

2. Cost or cost-effectiveness of all methods mentioned should be given in the text and mentioned in the summary part.

Added to Table 1, Table 2, and Summary. Costs are not openly published. Device cost is an issue raised in many editorials. Manufacturers/hospitals require confidentiality clauses within most sales contracts.

3. There should be a table giving the diagnostic yields, complication rates and cost / cost-effectiveness of all the methods mentioned

Seperate table has been incorporated.

4. The summary part should be re-written as an overview and comment on all mentioned in the review with a perspective of real life experience.

Summary has been edited to reflect experience and review items

Reviewer 2 Report

Sarkar et al wrote a detailed review on the latest landscape on navigational bronchoscopy. It is very informative and well written.

My suggestions:

(1) Line 67-68: Chicago cannot be the first centre to perform R-EBUS (It was the Japanese); also, the Ref 10 is probably wrong as it was about ENB by Tom Gildea. 

(2) Line 273- 274: As far as I know, ALL RB scopes are single-use / disposable (not just those form the Galaxy system). Please check

(3) For the session on tomosynthesis (the actual focus on the manuscript), please clarify for the audience whether the C-arm spin can be performed manually (as opposed to motorized) for each of the platforms.

(4) 1-2 pictorial illustration on fluoroscopic guidance from the tomosynthesis platform (e.g., LungVision) might help the audience understand better.

Author Response

(1) Line 67-68: Chicago cannot be the first centre to perform R-EBUS (It was the Japanese); also, the Ref 10 is probably wrong as it was about ENB by Tom Gildea. 

Now Line 69-70- Line is incorrect. Changed to reflect accurate discovery with appropriate reference

(2) Line 273- 274: As far as I know, ALL RB scopes are single-use / disposable (not just those form the Galaxy system). Please check

Correct- The Ion scope + Monarch scopes can be reused, but only 2-3 times. Line removed since all are disposible

(3) For the session on tomosynthesis (the actual focus on the manuscript), please clarify for the audience whether the C-arm spin can be performed manually (as opposed to motorized) for each of the platforms.

Changed lines 242-243, 262-263, 296 to clarify type of motion.

(4) 1-2 pictorial illustration on fluoroscopic guidance from the tomosynthesis platform (e.g., LungVision) might help the audience understand better.

Agreed. We reached out to the manufacturer for permission through multiple modalities. Unfortunately, they did not respond back to any of our correspondance.

Round 2

Reviewer 1 Report

The authors have revised the paper according to my comments.